

# Bioinformatics prediction and experimental verification of key biomarkers for diabetic kidney disease based on transcriptome sequencing in mice

Jing Zhao[1,2,*], Kaiying He[1,2,*], Hongxuan Du[1,2], Guohua Wei[2], Yuejia Wen[1,2], Jiaqi Wang[1], Xiaochun Zhou[2] and Jianqin Wang[2]

[1] Lanzhou University, Lanzhou, China
[2] Lanzhou University Second Hospital, Lanzhou, China
[*] These authors contributed equally to this work.

Corresponding authors
Xiaochun Zhou,
ery_zhouxc@lzu.edu.cn
Jianqin Wang,
ery_wangjqery@lzu.edu.cn

## ABSTRACT

**Background**. Diabetic kidney disease (DKD) is the leading cause of death in people with type 2 diabetes mellitus (T2DM). The main objective of this study is to find the potential biomarkers for DKD.

**Materials and Methods**. Two datasets (GSE86300 and GSE184836) retrieved from Gene Expression Omnibus (GEO) database were used, combined with our RNA sequencing (RNA-seq) results of DKD mice (C57 BLKS-32w db/db) and non-diabetic (db/m) mice for further analysis. After processing the expression matrix of the three sets of data using R software "Limma", differential expression analysis was performed. The significantly differentially expressed genes (DEGs) ($|logFC| > 1$, $p$-value $< 0.05$) were visualized by heatmaps and volcano plots respectively. Next, the co-expression genes expressed in the three groups of DEGs were obtained by constructing a Venn diagram. In addition, Gene Ontology (GO) and Kyoto Encyclopedia of Genes and Genomes (KEGG) pathway enrichment analysis were further analyzed the related functions and enrichment pathways of these co-expression genes. Then, qRT-PCR was used to verify the expression levels of co-expression genes in the kidney of DKD and control mice. Finally, protein-protein interaction network (PPI), GO, KEGG analysis and Pearson correlation test were performed on the experimentally validated genes, in order to clarify the possible mechanism of them in DKD.

**Results**. Our RNA-seq results identified a total of 125 DEGs, including 59 up-regulated and 66 down-regulated DEGs. At the same time, 183 up-regulated and 153 down-regulated DEGs were obtained in GEO database GSE86300, and 76 up-regulated and 117 down-regulated DEGs were obtained in GSE184836. Venn diagram showed that 13 co-expression DEGs among the three groups of DEGs. GO analysis showed that biological processes (BP) were mainly enriched inresponse to stilbenoid, response to fatty acid, response to nutrient, positive regulation of macrophage derived foam cell differentiation, triglyceride metabolic process. KEGG pathway analysis showed that the three major enriched pathways were cholesterol metabolism, drug metabolism– cytochrome P450, PPAR signaling pathway. After qRT-PCR validation, we obtained 11 genes that were significant differentially expressed in the kidney tissues of DKD mice
compared with control mice. (The mRNA expression levels of Aacs, Cpe, Cd36, Slc22a7, Slc1a4, Lpl, Cyp7b1, Akr1c14 and Apoh were declined, whereas Abcc4 and Gsta2 were elevated).

**Conclusion**. Our study, based on RNA-seq results, GEO databases and qRT-PCR, identified 11 significant dysregulated DEGs, which play an important role in lipid metabolism and the PPAR signaling pathway, which provide novel targets for diagnosis and treatment of DKD.

# INTRODUCTION

Diabetic mellitus (DM) is a chronic metabolic disease that seriously affects public health. According to the World Health Organization (WHO), about 629 million people will suffer from T2DM by 2045, and the complications may affect various organs throughout the body and have a high mortality and disability rate (*Tanase et al., 2020*). Diabetic kidney disease (DKD) is a common chronic complication of DM, which has gradually been the leading cause of death in patients with DM and end-stage renal disease (ESRD) (*Yang et al., 2020*). The main features of progression of DKD are hypertension, increased protein in urine, and decreased estimated glomerular filtration rate (eGFR). Early pathological changes mainly include thickening of the glomerular and tubular basement membrane, which gradually develops into glomerular extracellular matrix (ECM) accumulation and tubular interstitial fibrosis as the disease progresses, eventually causing irreversible damage to the renal structure. Most experts believe that the pathogenesis of DKD is mainly due to the interaction of renin-angiotensin-aldosterone system (RAAS), advanced glycation end products (AGEs), transforming growth factor-$\beta$1 (TGF-$\beta$1), protein kinase C (PKC), mitogen-activated protein kinases (MAPKs) and reactive oxygen species (ROS), which affect renal function through inflammation and oxidative stress (*Samsu, 2021*). Although various treatments have been used to improve metabolism, hemodynamic disturbances, and fibrosis, the mortality rate of patients with DKD remains high. Therefore, the key to improving the quality of life and survival rate is to find more useful biomarkers for diagnosis of DKD and to develop new strategies and prevention of deterioration of renal function.

In recent years, many biomarker molecules have been found to be associated with changes in renal structure and function in DKD patients, such as urine markers, serum/plasma markers, *etc.* (*Colhoun & Marcovecchio, 2018*). With the development of the new generation of high-throughput sequencing technology and bioinformatics techniques, the ability of humans to understand diseases from the root has greatly improved, and more and more disease-related risk genes have been discovered. This promises revolutionary advances in disease diagnosis and treatment in the future (*Rego & Snyder, 2019*). Many microarray-based studies have shown that non-coding RNAs, mRNAs and the protein it encodes

play important roles in the pathogenesis of DKD. They influence disease occurrence, progression, and prognosis through their interactions and regulation of signaling pathways. For example, studies have shown that leucine rich- $\alpha$ -2 glycoprotein 1 (LRG1) expression is increased in kidneys of diabetic patients and mice, it can be used as a biomarker for diagnosis of DKD (*Hong et al., 2019*). The expression of SH3YL1 in the serum of DKD patients, in kidney tissues of db/db mice, and in podocytes, is higher than control group, so it also has the potential as a biomarker for the diagnosis of DKD (*Choi et al., 2021*). Some nephrologists have used bioinformatics methods to analyze the datasets uploaded by other authors in GEO database, and screened out candidate genes in DEGs through functional analysis and protein interaction network construction, which is considered as a possible biomarker of DKD (*Gao et al., 2021*). The pathogenesis of DKD is complex, which is related to oxidative stress, inflammation, autophagy, apoptosis, and other mechanisms. However, its pathogenesis still needs further study. Although many mRNAs and proteins they encode are currently thought to play a role in DKD, there are few consistent results across studies. We need to study by using the method of biological technology, such as through such as single cell sequencing, high throughput sequencing, multi-omics combined analysis to obtain more data analysis, or we share our own raw data in the database so that more researchers can study it. Then, some basic experiments are added to verify the mechanism of genes action in diseases, such as qRT-PCR and Western blot, so that the research results on the mechanism of genes in DKD can have higher authenticity and reliability.

Unlike most bioinformatics studies, we added our own sequencing data, and performed a comprehensive analysis with data similar to our sequencing uploaded by other authors in the GEO database. The candidate gene was verified by qRT-PCR again, increasing the accuracy of the study. Our analysis was conducted following the procedure presented in Fig. 1. Finally, 13 co-expression genes were determined by the combined analysis of the 3 datasets, qRT-PCR verified that 11 of them had the same expression trend as the above sequencing results and analysis showed several of these genes may affect the occurrence of disease through lipid metabolism and the PPAR signaling pathway.

## MATERIALS AND METHODS

### Data source

#### *Transcriptome sequencing of kidney tissues from DKD and Control mice*

**(1) Animal model**

Twelve C57 BLKS-db/db mice and twelve db/m mice(6-week-old, male)were selected as the model group (body weight 34.35 $\pm$ 2.52 g),normal control group(body weight 18.94 $\pm$ 1.44 g) respectively. They were purchased from Nanjing Institute of Model Zoology and reared in the barrier system of Animal Experimental Research Centre of Lanzhou University Second Hospital. The feeding temperature was (20 $\pm$ 2) °C, the humidity was 40%–70%, the light alternated between light and dark every 12 h, the ordinary Specific Pathogen Free (SPF) food was fed and drank water freely. Starting from the 8th week, the body weight, blood sugar (Sinocare Inc), and 16-hour urine microalbumin (Enzyme Linked Immunosorbent Assay (ELISA) kit; ML Bio, Charlotte, NC, USA) of the mice were measured every 4 weeks, the blood was collected from the tail vein. We obtained blood

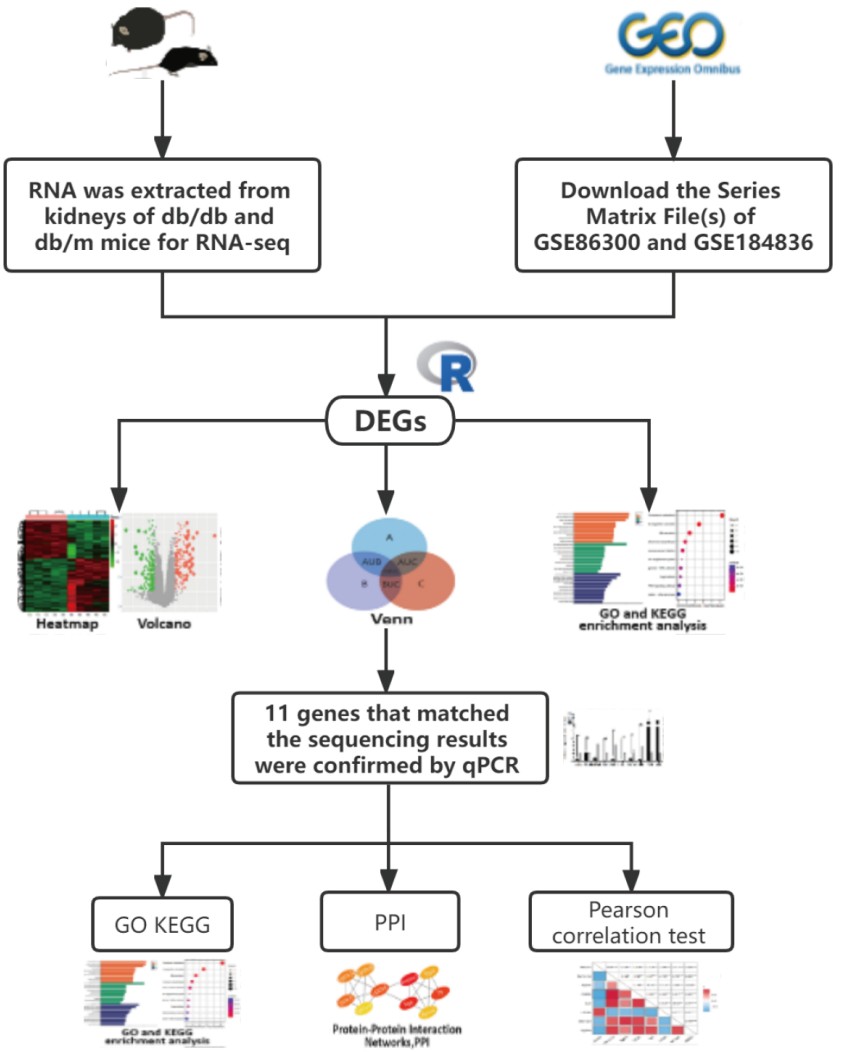

**Figure 1** The workflow of this study.

from the heart at the 8th and 32nd weeks (six mice were randomly selected for blood collection from heart at week 8, and the other six were fed to 32 weeks for blood collection in heart), stored at 4 °C for 1 h, and centrifuged for 10 min (3000 rpm, 4 °C) to obtain the serum. Then the serum creatinine (Scr), blood urea nitrogen (BUN), total cholesterol (TC), triglycerides (TG), low-density lipoprotein cholesterol (LDL-C), high-density lipoprotein cholesterol (HDL-C) were measured by ELISA kit (Nanjing Jiancheng Bioengineering Institute). We fed the mice with a normal diet for 26 weeks, then the mice were euthanized by intraperitoneal injection of 10% chloral hydrate and the kidney tissues were collected. 50 mg of the right kidney was divided into cryogenic vials, frozen with liquid nitrogen and stored in the refrigerator at −80 °C. All the experiments involving animals were performed after Lanzhou University Second Hospital Institutional Ethical Committee's approval

and under the strict adherence to the National Institutes of Health Guide for laboratory animals' care and use, our ethical review acceptance number is D2019-198.

**(2) RNA extraction from kidney tissues**

Approximately 50 mg of each kidney tissue sample was placed in grinding tubes, and 1 ml of Trizol was added to each tube and ground completely. Then the samples were allowed to stand for 5 min at room temperature, and 0.2 ml of chloroform was added to each tube and shaken vigorously for 15 s. After the samples stand at room temperature for 3 min, the supernatant was centrifuged at high speed (12,000 rpm, 4 °C) for 15 min and transferred to a Rnase-free centrifuge tube. Next, add 0.5 mL isopropyl alcohol, mix gently and let stand at room temperature for 10 min, centrifuge for 10 min (12,000 rpm, 4 °C), wash the precipitate with one mL 75% ethanol, centrifuge for 5 min (7,500 rpm, 4 °C), remove the liquid, dry at room temperature for 10 min. 150 uL diethyl pyrocarbonate (DEPC) $H_2O$ was added and gently mix. At last, the precipitate was left at 55 °C for 10 min to dissolve and stored temporarily at −80 °C for further sequencing.

**(3) RNA-sequencing (RNA-seq)**

In this study, we need to consider and deal with the difference expression caused by the biological variability, by far the most commonly used and most effective way is to set up biological replic validates. All biological repeat samples under the same conditions were extracted and built with the same person and batch, sequenced with same Run and Lane, and conducted a detailed analysis of the abnormal samples. Meanwhile, we set the Power value as 0.9, $\alpha$ value as 0.05 and then used the R language RNAseqPower package to calculate the sample size was 2.44, which could achieve 1.5 times of change, proving that the number of mice in each group selected as 3 is valid in our experiment.

Illumina's high-throughput sequencing platform was used to perform transcriptome sequencing on kidney tissue samples from DKD mice and Control mice. Clean data were obtained by filtering data from the Illumina high-throughput sequencing platform sequenced with the indicated reference genome. Next, the sequencing data were obtained through sequence structure analysis, library quality assessment and comparing with the reference genome. Finally, the number of mapped reads and transcript length were normalized in the sample. Fragments Per Kilobase Million (FPKM) was used as an indicator to measure gene expression levels. The expression matrix of FPKM can be obtained by computational formula shown as: FPKM = (cDNA Fragments\over (Mapped Fragments (Millions)*Transcript Length(kb))).

### GEO database

GEO database was built by the National Center for Biotechnology Information (NCBI) and is a gene expression database and online genome resource that collects high-throughput gene expression data uploaded from research institutes around the world. Diabetic nephropathy or diabetic kidney disease was entered as search objects. Gene expression microarray datasets GSE86300 and GSE184836 were selected and downloaded. The criteria for selecting the datasets were as follows: DKD and Control mice models with detailed gene expression information. The GSE86300 dataset, based on the GPL7546 platform, includes five DKD kidney tissue samples and five Control kidney tissue samples. The GSE184836

dataset, based on the GPL21103 platform, includes three DKD kidney tissue samples and three Control kidney tissue samples. The detailed information on these microarray datasets is listed in Table S1.

## Data processing methods
### Differential expression analysis
All differential analyses of the 3 datasets were performed using the Limma packages in R/Bioconductor software. The script for the Limma package is available in Supplemental File S2. Genes in DKD renal tissue were up- or down-regulated compared with control groups, $p$-value less than 0.05 and |$\log_2$ fold change (FC)| greater than 1 were considered statistically significant in differential analysis of the 3 datasets, log2 FC >1 was regarded as up-regulated genes and log2 FC < $-1$ was down-regulated.

### Volcano plots and heatmaps
Significant DEGs (|log2 FC| >1, $p$-value < 0.05) were visualized using heatmaps and volcano curves. Both visualizations were completed by http://www.bioinformatics.com.cn, an online platform for data analysis and visualization. We drew heatmaps by importing FPKM values of the two groups to the website, and using the same method, we imported gene names, log2 FC values, and $P$-value to map the volcano.

### Venn diagram
Determine the common genes of 3 datasets by creating a Venn diagram. The Venn diagram is completed on the "Weishengxin" website. We imported three sets of datasets to the Venn diagram tool on the website, draw Venn diagrams, and obtain the intersection.

### Functional enrichment analysis
Gene Ontology (GO) annotation and Kyoto Encyclopedia of Genes and Genomes (KEGG) pathway enrichment analysis were performed for DEGs. GO Annotation included analysis of biological processes (BP), cell components (CC), and molecular functions (MF). KEGG is an online database for pathway analysis of a large amount of genetic information. GO functional annotation analysis and KEGG pathway enrichment analysis were performed in the website "Weishengxin", import our gene name and log2 FC into the website for enrichment analysis. The significant enrichment threshold was set as $p$-value < 0.05, ten functions and pathways with the lowest $p$-values were selected as the top 10.

### Quantitative real-time polymerase chain reaction(qRT-PCR)
Total RNA was extracted from fresh mice kidneys of DKD and control groups using the TRIZOL method. Total RNA (1 ug) was transcribed into cDNA using the GoScript$^{TM}$ Reverse Transcription System according to the manufacturer's protocol (Promega). qRT-PCR was performed on the ABI7500 system using the GoTaq® qPCR Master Mix (Promega). All data were normalized to $\beta$-actin expression. Relative RNA expression was calculated using the $2^{-\Delta\Delta CT}$ method. Detailed information of primers is listed in Table S2.

*PPI network analysis and Pearson correlation test*

The protein-protein interaction (PPI) network was created from the "STRING" database for target genes and is designed to discover the interaction of target genes and their interactions with other proteins, and Cytoscape (v3.9.0) was used to edit the graphics. The expression matrix of the 3 datasets was merged after removing batch effect by the Sangerbox tool, and the Pearson correlation test was adopted for evaluation of the interactions between DKD related genes at the mRNA level using R.

## Statistical analysis

All values are expressed as the mean $\pm$ SEM. Statistical analysis was performed using the statistical package SPSS for Windows Version 7.51 (SPSS, Inc., Chicago, IL, USA) and GraphPad Prism 9. Results were analyzed using Student t test for multiple comparisons. In our qRT-PCR experiment, each sample was added with three holes each time, and three groups of different samples were repeated for verification. Finally, we performed T test on the $2^{-\Delta\Delta CT}$ of the three groups of data to obtain the *P*-value. Statistical significance was detected at the 0.05 level.

## RESULTS

### General condition and biochemical indexes of mice

From the age of 8 weeks, compared with db/m mice, the water and food intake of db/db mice increased gradually, the body weight increased ($p < 0.05$), blood sugar increased ($p < 0.05$), urinary protein excretion rate and various biochemical indexes changed significantly ($p < 0.05$) (Fig. 2).

### Differential expression genes

Based on our RNA-seq results, a total of 125 differentially expressed mRNAs were identified, including 59 up-regulated genes and 66 down-regulated genes. After screening the genes with differential expression in GSE86300, 336 genes with differential expression were determined, including 183 up-regulated genes and 153 down-regulated genes. In GSE184836 DEGs, there were 315 DEGs, including 76 up-regulated genes and 117 down-regulated genes. The 3 datasets were analyzed according to the criteria |log$_2$ FC| >1 and *p*-value < 0.05. To visualize the DEGs, we constructed heatmaps (Figs. 3A, 3B and 3C) and volcano curves (Figs. 3D, 3E and 3F) and then created a Venn diagram to obtain 13 co-expression genes among the 3 datasets (Fig. 3G).

### Enrichment analysis

GO functional annotation and KEGG pathway enrichment analysis were performed for the DEGs from the three datasets. The GO analysis results of our RNA-seq showed that the BP analysis was mainly enriched in the organic anion transport, fatty acid metabolic process, organic hydroxy compound metabolic process, steroid metabolic process, response to stilbenoid, while CC is mainly concentrated in chylomicron, very-low-density lipoprotein particle, triglyceride-rich plasma lipoprotein particle, plasma lipoprotein particle, lipoprotein particle. The top 5 of MF are monooxygenase activity, oxidoreductase

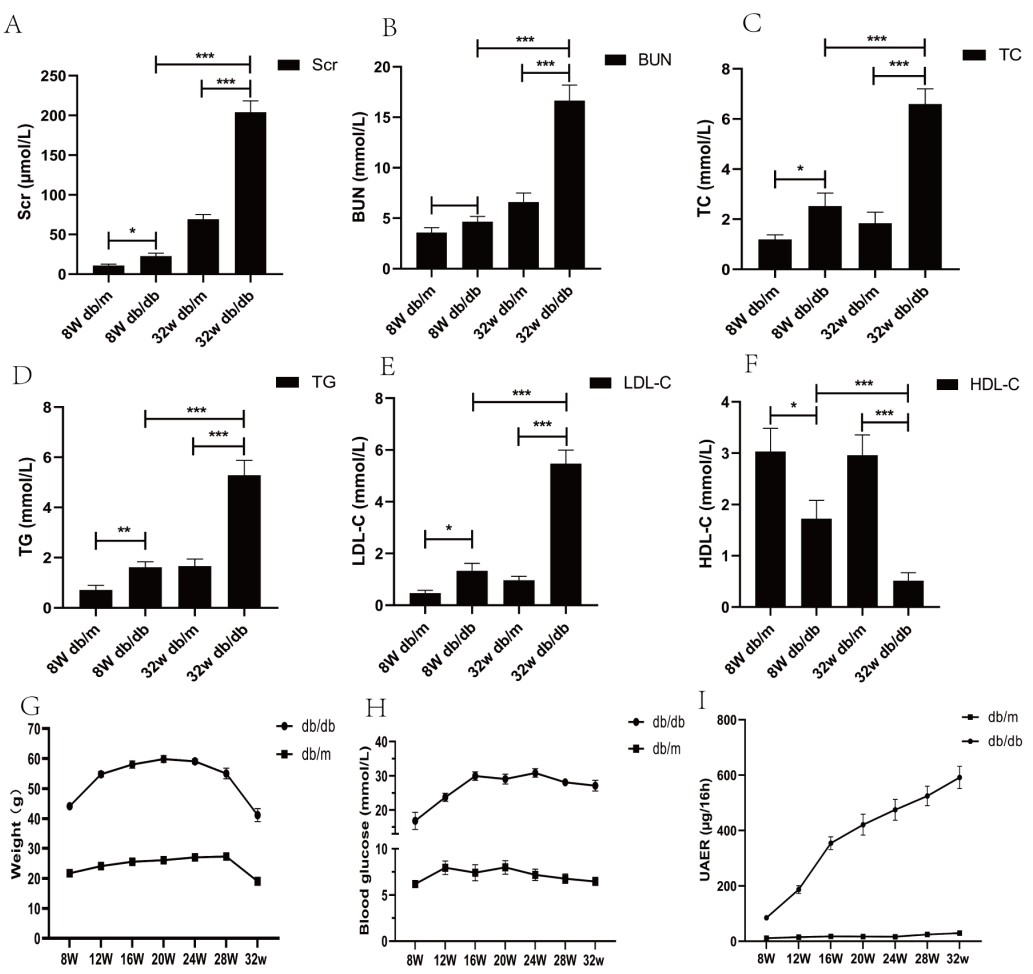

**Figure 2** **Biochemical parameters of blood and urine in DKD mice model of different week ages.** (A) Comparison of serum creatinine in mice; (B) comparison of blood urea nitrogen in mice; (C) comparison of total cholesterol in mice; (D) comparison of triglycerides in mice; (E) comparison of low-density lipoprotein in mice; (F) comparison of high-density lipoprotein in mice; (G) weight comparison of mice; (H) comparison of blood glucose in mice; (I) comparison of urinary albumin excretion rate (UAER) in mice. * $p < 0.05$, ** $p < 0.01$, *** $p < 0.001$.

activity, organic anion transmembrane transporter activity, lipid transporter activity, glucuronosyltransferase activity (Fig. 4A). KEGG pathway analysis showed that the top three enrichment pathways of the DEGs are bile secretion, steroid hormone biosynthesis, drug metabolism—other enzymes (Fig. 4B).

GSE86300 analysis showed that the BP of DEGs was mainly in the negative regulation of blood coagulation, negative regulation of hemostasis, fibrinolysis (Fig. 4C). At the same time, KEGG analysis showed they were mainly in the Cholesterol metabolism, Complement and coagulation cascades, butanoate metabolism, PPAR signaling pathway (Fig. 4D). GSE184836 analysis indicated that organic anion transport, organic hydroxy compound metabolic process, steroid metabolic process as the top three BP (Fig. 4E). Meanwhile, KEGG analysis manifested that metabolism of xenobiotics by cytochrome

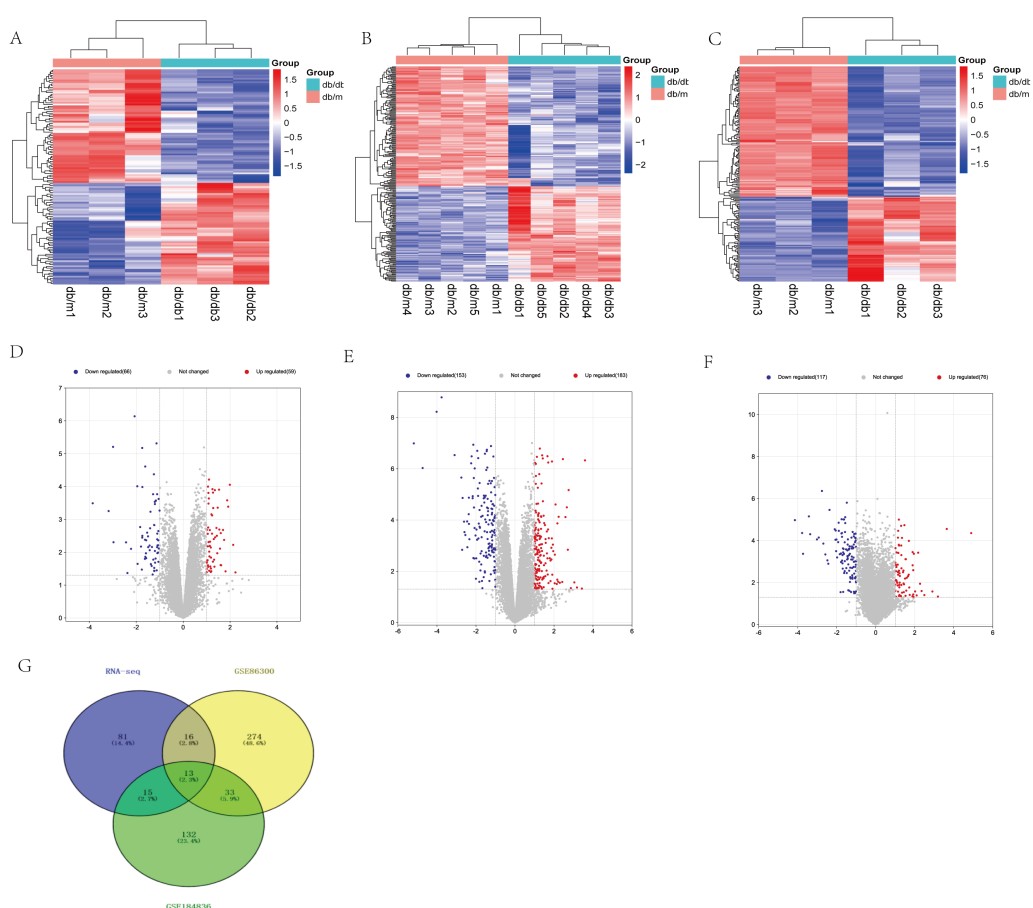

**Figure 3   Differential expression analysis of our RNA-seq and two GEO datasets (GSE86300 and GSE184836).** (A) Heatmap of DEGs in our RNA-seq. (B) Volcano map of our RNA-seq. A total of 59 upregulated and 66 downregulated DEGs were identified between the DKD and the Control group. (C) Heatmap of DEGs in GSE86300. (D) Volcano map of DEGs in GSE86300. A total of 183 up-regulated and 153 down-regulated DEGs were identified between the DKD and the Control group. (E) Heatmap of DEGs in GSE184836. (F) Volcano map of DEGs in GSE184836. A total of 76 up-regulated and 117 down-regulated DEGs were identified between the DKD and the control group. (G) Venn diagram of three DEGs groups. A total of 13 co-expression genes were obtained. Volcano map exhibit significantly differentially expressed genes, in volcano map, red bubbles mean up-regulated genes, blue bubbles mean down-regulated genes, and gray bubbles mean non-significant genes. The dots in the area above the horizontal dotted line have a *P*-value < 0.05. The dots outside the two vertical dotted lines have a |log2FC| > 1. Based on gene expression matrix, clustering analysis was shown in heatmap, in heatmap, red mean up-regulated genes, blue mean down-regulated genes. (|log2 FC| >1 and *p*-value<0.05).

P450, cholesterol metabolism, butanoate metabolism as the top three (Fig. 4F). The results of GO and KEGG analysis are sorted by *p*-value.

## Venn diagram

Thirteen overlapping genes, namely, Aacs, Cpe, Cd36, Slc22a7, Slc1a4, Lpl, Kynu, Cyp7b1, Akr1c14, Apoh, Fmo5, Abcc4, Gsta2 were identified from the three datasets by creating a Venn diagram. The expression levels and trends of the above 13 genes in three databases were listed in Table 1.

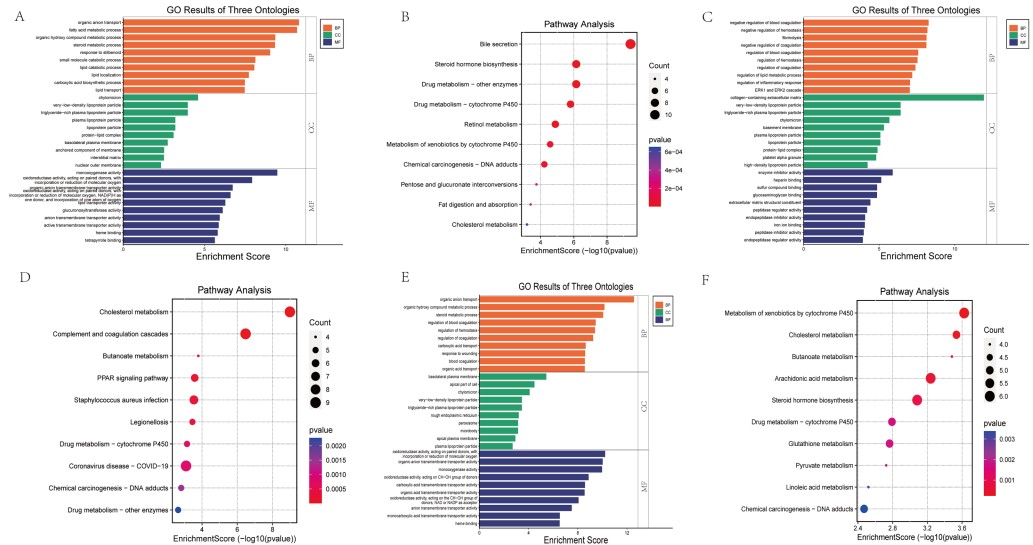

**Figure 4** **Bar graph of GO Annotation and dot plot of KEGG pathway enrichment analysis of DEGs.**
(A, B) Our RNA-seq (Top 10). (C, D) GSE86300 (Top 10). (E, F) GSE184836 (Top 10). Bar graph shows
that DEGs of the three groups are enriched in several biological processes (BP), cell components (CC),
molecular functions (MF). In the bar graph, we sorted the top 10 of BP, CC and MF by *p*-value and visu-
alize them. In the dot plot, the color represents the *p*-value, and the size of the spots represents the gene
number.

## Identification of 13 overlapping genes expression by qRT-PCR

The mRNA levels of the 13 overlapping genes were verified by qRT-PCR in DKD and
control mice kidney tissues, meanwhile, the results indicated that the mRNA levels of
Aacs, Cpe, Cd36, Slc22a7, Slc1a4, Lpl, Cyp7b1, Akr1c14, Apoh were down-regulated, while
Abcc4 and Gsta2 were up-regulated, these changes were statistically significant ($p < 0.05$).
In addition, qRT-PCR results of Fmo5 showed that its expression was up-regulated in
DKD kidney tissues, which is contrary to the sequencing results, the expression of Kynu
showed no statistical significance in the DKD group compared with the Control ($p > 0.05$)
(Fig. 5). In conclusion, the mRNA expression levels of these 11 genes verified by qRT-PCR
were consistent with the sequencing results (Aacs, Cpe, Cd36, Slc22a7, Slc1a4, Lpl, Cyp7b1,
Akr1c14, Apoh, Abcc4, Gsta2).

## Protein interaction analysis

The "STRING" database was used for PPI analysis. The results of PPI analysis of the above
verified 11 genes were similar to those of functional annotation of GO and KEGG pathway
enrichment analysis (Figs. 6A–6C). These genes apparently showed mainly correlation with
lipid metabolism, PPAR signaling pathway, and of these 11 genes, three had an obvious
interaction relationship, namely Cd36, Apoh, and Lpl (Fig. 6C). We also obtained some
genes associated with them (Fig. 6D). Similarly, the enrichment analysis of the BP of 11
genes revealed that the above three genes were all enriched in the triglyceride metabolic
process. Additional information of the 11 verified genes related to BP analysis is listed in

**Table 1    Expression of 13 overlapping genes in three datasets.**

| group | RNA-seq | | GSE86300 | | GSE184836 | | |
|---|---|---|---|---|---|---|---|
| Gene | logFC | *p* value | logFC | *p* value | logFC | *p* value | Up/down |
| Aacs | −2.0679 | 0.0000 | −1.3349 | 0.0016 | −1.5640 | 0.0000 | Down |
| Abcc4 | 1.1880 | 0.0145 | 1.4252 | 0.0000 | 1.3084 | 0.0002 | Up |
| Akr1c14 | −2.4325 | 0.0050 | −1.8565 | 0.0001 | −3.4254 | 0.0000 | Down |
| Apoh | −1.4246 | 0.0087 | −1.5422 | 0.0112 | −1.1246 | 0.0002 | Down |
| Cd36 | −1.1073 | 0.0002 | −2.1099 | 0.0000 | −2.3700 | 0.0000 | Down |
| Cpe | −1.7440 | 0.0000 | −1.4616 | 0.0025 | −2.0109 | 0.0001 | Down |
| Cyp7b1 | −2.9671 | 0.0050 | −2.5894 | 0.0003 | −3.0130 | 0.0001 | Down |
| Fmo5 | −1.5849 | 0.0108 | −2.3630 | 0.0000 | −1.4042 | 0.0014 | Down |
| Gsta2 | 1.1099 | 0.0364 | 1.0850 | 0.0106 | 3.6420 | 0.0000 | Up |
| Kynu | 1.0834 | 0.0049 | 1.8903 | 0.0005 | 1.4218 | 0.0027 | Up |
| Lpl | −1.0558 | 0.0023 | −1.0108 | 0.0014 | −1.6906 | 0.0022 | Down |
| Slc1a4 | −1.2027 | 0.0003 | −1.2780 | 0.0010 | −1.4501 | 0.0005 | Down |
| Slc22a7 | −3.8535 | 0.0003 | −5.1868 | 0.0000 | −4.1435 | 0.0000 | Down |

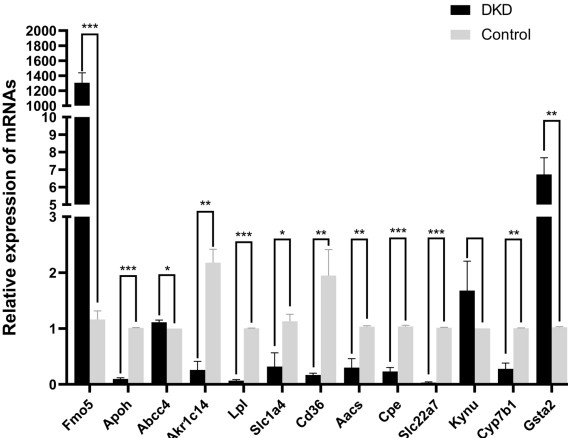

**Figure 5    The relative mRNA expression of 13 overlapping genes in DKD and control group determined by qRT-PCR.** * $p < 0.05$, ** $p < 0.01$, *** $p < 0.001$.

Table 2. In addition, through the Pearson correlation test, there were prominently positive or negative correlations between the DKD related genes at the mRNA levels (Fig. 6E).

# DISCUSSION

In recent years, the consequences of DKD have gradually attracted attention. With the increasing incidence, it brings a heavy medical burden to society. Nowadays, the clinical diagnosis of DKD mainly depends on the protein content in urine, and the treatment aims to reduce the protein content in urine. With the development of sequencing technology, we now realize that genes play a key role in the diagnosis, occurrence, progression and
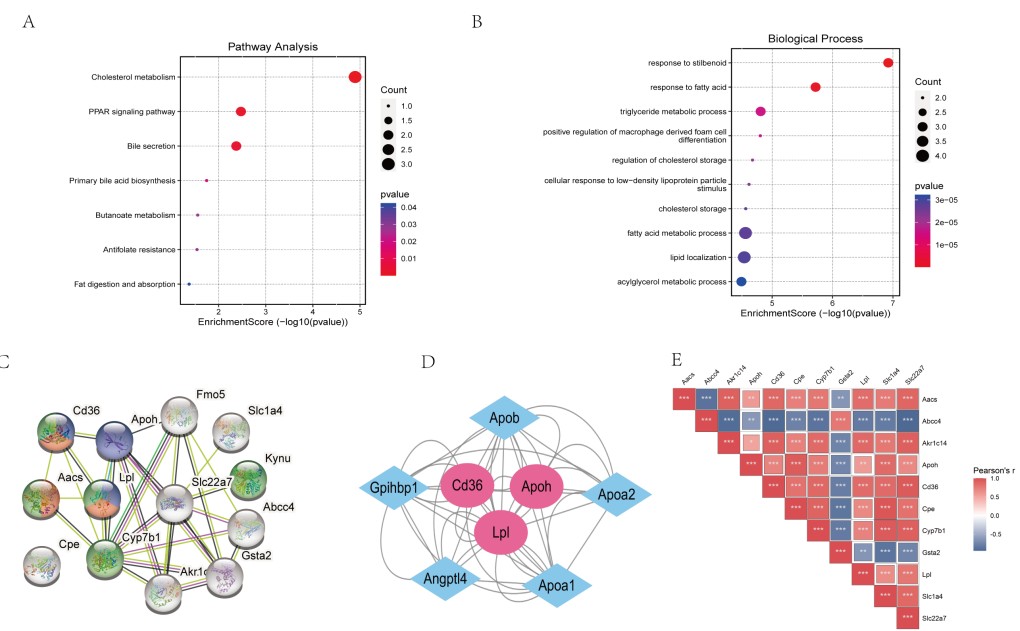

**Figure 6** **Functional analysis of key genes and their interactions.** (A) Dot plot of KEGG pathway enrichment analysis of verified 11 genes, in the dot plot, the color represents the *p*-value, and the size of the spots represents the gene number. (B) Bar plot of GO enrichment analysis of verified 11 genes. (C) PPI network of verified 11 DEGs, which also showed that other genes were involved. In the PPI analysis, green represents the monocarboxylic acid metabolic process, red represents the PPAR signaling pathway and blue represents the lipid metabolic process, white has no meaning. (D) The protein networks of the three key genes and several related genes. (E) Pearson correlation test between 11 DKD related genes, blue represents negative correlation, while red represents positive correlation.

**Table 2** **The BP analysis of 11 verified genes.**

| ID | Description | *P* value | Gene |
|---|---|---|---|
| GO:0035634 | Response to stilbenoid | 1.20E−07 | Cd36/Slc22a7/Gsta2 |
| GO:0070542 | Response to fatty acid | 1.91E−06 | Aacs/Cd36/Lpl |
| GO:0006641 | Triglyceride metabolic process | 1.55E−05 | Cd36/Lpl/Apoh |
| GO:0010744 | Positive regulation of macrophage derived foam cell differentiation | 1.57E−05 | Cd36/Lpl |
| GO:0010885 | Regulation of cholesterol storage | 2.12E−05 | Cd36/Lpl |
| GO:0071404 | Cellular response to low-density lipoprotein particle stimulus | 2.42E−05 | Cd36/Lpl |
| GO:0010878 | Cholesterol storage | 2.74E−05 | Cd36/Lpl |
| GO:0006631 | Fatty acid metabolic process | 2.76E−05 | Aacs/Cd36/Lpl/Akr1c14 |
| GO:0010876 | Lipid localization | 2.90E−05 | Cd36/Lpl/Apoh/Abcc4 |
| GO:0006639 | Acylglycerol metabolic process | 3.23E−05 | Cd36/Lpl/Apoh |

treatment of DKD. Thus, many mRNAs and non-coding RNAs associated with DKD have been discovered and supported by a large number of studies.

In our study, the significant increase of UAER, as well as the changes of Scr and other biochemical indicators, could indicate the decline of renal function of mice. We found 11 genes whose expression levels were significantly altered in the kidney tissues of DKD mice. Aacs, Cpe, Cd36, Slc22a7, Slc1a4, Lpl, Cyp7b1, Akr1c14, Apoh were down-regulated whereas the expression levels of Abcc4 and Gsta2 were upregulated. We found that these genes are homologous genes in humans and mice in the National Center for Biotechnology Information (NCBI) database, and the genes have been studied in human and mice models of different diseases. Therefore, we have reason to believe that these genes have similar functions in human disease and mice model research, so we can screen important genes on the basis of mice study, and provide some reference data for the study of this disease in human.

In our findings, what attracted our attention was that some genes were involved in lipid metabolism and PPAR signaling pathway. It is well known that glucose and lipid metabolism disorder has become a very common metabolic defect, lipid metabolism disorders are closely associated with renal dysfunction. Researchers have confirmed that dyslipidemia and the increased free fatty acids (FFA) are risk factors for insulin resistance and glucolipid toxicity is an important cause of T2DM (*Lytrivi et al., 2020*). Lipid toxicity is mainly involved in renal damage related to lipid toxicity through the activation of inflammation, oxidative stress, mitochondrial dysfunction and apoptosis (*Opazo-Ríos et al., 2020*), and increased lipophagy ameliorates renal damage, lipid deposition, oxidative stress and apoptosis in the kidney (*Han et al., 2021*). In addition, PPAR signaling pathway has been proved to be significantly related to lipid metabolism by a large number of studies, and its role in DKD has also been widely recognized (*Kim et al., 2018*; *Mao et al., 2021*).

The results of PPI showed that five genes were enriched in lipid metabolic process, namely Cd36, Lpl, Apoh, Aacs and Cyp7b1, at the same time, Cd36, Lpl and Apoh were related to each other, which may act together to influence the occurrence and development of DKD. We found that these genes have been extensively reported in the literature in past studies. For example, some studies suggested that Cd36 is involved in fatty acid uptake, apoptosis, angiogenesis, phagocytosis, inflammation and atherosclerosis (*Pepino et al., 2014*). Prior data indicated that high glucose exacerbates fatty acid uptake and deposition through increased expression of Cd36 *via* the AKT-PPAR $\gamma$ pathway (*Alkhatatbeh et al., 2013*; *Mitrofanova et al., 2020*). Cd36 knockdown prevents renal tubular injury, tubulointerstitial inflammation, and oxidative stress in 16 weeks db/db mice (*Hou et al., 2021*). On the contrary, the loss of Cd36 in mice leads to phenotypes such as lymphatic drainage, visceral fat, and glucose intolerance, which increases the risk for T2DM (*Cifarelli et al., 2021*). In our study, compared with the control mice, Cd36 in the DKD group was down-regulated. We speculated the reason might be that our mice had reached the late stage at the age of 32 weeks, so the expression level of DKD was different from that in the early stage. In addition, studies have shown that in C57BL/6J mice, Lpl was mainly expressed intracellular and restricted to the proximal tubule (*Nyrén et al., 2019*). In the DKD group, renal Lpl mRNA expression was significantly decreased, and it increased the level of triglyceride

(TG) in renal tissue (*Herman-Edelstein et al., 2014*). PPAR is involved in lipid metabolism and it regulates the activity of proteins like LPL (*La Paglia et al., 2017*). Apoh is involved in a variety of physiological processes, including lipoprotein metabolism, coagulation, and antiphospholipid autoantibody production (*Athanasiadis et al., 2013*). It is also associated with BMI in diabetics and cardiovascular risk factors and is considered an anti-obesity factor (*Hasstedt et al., 2016*). In addition to these, Aacs has been widely recognized for its role in cellular response to glucose stimulation, regulation of insulin secretion, and fatty acid metabolism. Previous studies have shown that 80% inhibition of Aacs can partially inhibit glucose-induced insulin release (*MacDonald et al., 2007*). Besides, the enzyme encoded by Aacs is a ketone body-utilizing ligase with a role in lipid synthesis through the non-oxydative pathway (*Haydar et al., 2019*), knockdown of Aacs *in vivo* resulted in the reduction of total blood cholesterol (*Hasegawa et al., 2012*). Studies have shown that Cyp7b1 is related to the synthesis of bile acid (BA) (*Evangelakos et al., 2022*), which is the final product of cholesterol catabolism and is related to the regulation of lipid, glucose and energy metabolism, inflammation, detoxification of drug metabolism (*Li & Chiang, 2014*), as well as the development of liver steatosis and inflammation (*Evangelakos et al., 2022*).

In addition, some genes are not enriched in lipid metabolism and PPAR pathways, and more studies are needed in the future to fill in the gaps in their functional studies. For example, Akr1c14, a metabolic enzyme which has been thought to play a role in the maintenance of normal adipocyte metabolism (*Becker et al., 2019*). Abcc4 is highly expressed in the kidney of mice, which has been verified that the expression of Mrp4/Abcc4 increased in liver and kidney of obese and diabetic patients (*Donepudi et al., 2021*). Carboxy peptidase E (Cpe), an enzyme that converts the pro-insulin to insulin, which suggest that Cpe may be related to the occurrence of T2DM (*Sabiha et al., 2021*), rescuing proinsulinmia caused by reduced CPE may be a new approach to treat early diabetes (*Jo, Lockridge & Alejandro, 2019*). Nuclear factor erythroid 2-related factor 2 (Nrf2) is known to regulate cellular oxidative stress and induce expression of antioxidant genes. Gsta2 is a downstream signaling molecule of Nrf2 and also participates in the antioxidant stress effect of Nrf2 (*Kim et al., 2015*), which was researched in the mice with surgically induced Unilateral ureteral obstruction (UUO). Slc22a7 regulates the absorption, distribution, and excretion of a wide variety of environmental toxins and clinically important drugs as an organic anion transporter (*Zhou & You, 2007*). Slc1a4 is highly expressed in the central nervous system, and relevant studies are still lacking.

The results of PPI suggest that our key genes are not only related to lipid metabolism and PPAR pathways, but also have other related genes. For example, they interact with Angptl4, Apoa1, Apoa2 and Apo b. Angptl4 has been studied as a potent inhibitor of Lpl that regulates cellular uptake of triglycerides and promotes fatty acid oxidation (*Zhou et al., 2021*). *Aryal et al. (2016)* found that the expression of Cd36 increased in Angptl4$^{-/-}$ macrophages. Apo a and Apo b are mainly studied in obesity and cardiovascular diseases, studies suggest that Apoa1, Apo b, and Apo b/Apoa1 ratio have been regarded as the predictors of microvascular and macrovascular complications of diabetes (*Moosaie et al., 2020*). These genes are concentrated in PPAR signaling pathway, which is thought to be involved in the regulation of glucose and lipid metabolism, endothelial function and

inflammation (*Wang, Dougherty & Danner, 2016*). These genes may also have implications in the influence of key genes on disease. Although we have seen less research of these genes on kidney disease, we speculate from our current study that it may play an important role in kidney disease, more experiments will investigate the mechanism in the future.

However, the present study has certain limitation. First, we determined the renal function status of mice by detecting the hematuria biochemical indexes of the mice, but there was no corresponding pathological evidence. In addition, qRT-PCR was simply used to verify their mRNA expression levels in the kidney tissues of DKD mice, while Western blot was not used to verify their protein expression level, so the target genes should be further verified in experimental studies. The present results were obtained using a bioinformatics screening to identify several DEGs between DKD and control groups, the results showed that several genes not only played an important role in the lipid metabolism of DKD, but also had a close relationship with PPAR signaling pathway. Importantly, the information provided in this study was not limited to the 11 verified genes, but perhaps included certain other typical DEGs. The current results provide a worthy resource for future research on DKD.

## CONCLUSION

In conclusion, our study, based on RNA-seq results, the GEO database and qRT-PCR, identified 11 significant dysregulated DEGs, which play an important role in lipid metabolism and the PPAR signaling pathway, which provide novel targets for diagnosis and treatment of DKD, additional basic and clinical studies are needed to further validate these targets.

### Funding

This work was supported by the National Natural Science Foundation of China (No. 81960142), Youth Science and Technology Fund Program of Gansu Province (No. 21JR1RA157), Talent Innovation and Entrepreneurship Project of Lanzhou City, Gansu Province (2021-RC-94), Lanzhou Chengguan District Talent Entrepreneurship and Innovation Project (2021RCCX0027), Lanzhou University Second Hospital Youth Fund (CY2021-QN-B01), and Project of Department of Education of Gansu Province (2022B-050). Meanwhile, our experiments are supported by the Clinical Medical Research Center of Gansu Province (21JR7RA436). There was no additional external funding received for this study. The funders had no role in study design, data collection and analysis, decision to publish, or preparation of the manuscript.

### Grant Disclosures

The following grant information was disclosed by the authors:
National Natural Science Foundation of China: 81960142.
Youth Science and Technology Fund Program of Gansu Province: 21JR1RA157.
Talent Innovation and Entrepreneurship Project of Lanzhou City, Gansu Province: 2021-RC-94.

Lanzhou Chengguan District Talent Entrepreneurship and Innovation Project: 2021RCCX0027.
Lanzhou University Second Hospital Youth Fund: CY2021-QN-B01.
Project of Department of Education of Gansu Province: 2022B-050.
Clinical Medical Research Center of Gansu Province: 21JR7RA436.

## Competing Interests

The authors declare there are no competing interests.

## Author Contributions

- Jing Zhao performed the experiments, authored or reviewed drafts of the article, and approved the final draft.
- Kaiying He performed the experiments, authored or reviewed drafts of the article, and approved the final draft.
- Hongxuan Du performed the experiments, prepared figures and/or tables, and approved the final draft.
- Guohua Wei analyzed the data, prepared figures and/or tables, and approved the final draft.
- Yuejia Wen analyzed the data, prepared figures and/or tables, and approved the final draft.
- Jiaqi Wang analyzed the data, prepared figures and/or tables, and approved the final draft.
- Xiaochun Zhou conceived and designed the experiments, authored or reviewed drafts of the article, and approved the final draft.
- Jianqin Wang conceived and designed the experiments, authored or reviewed drafts of the article, and approved the final draft.

## Animal Ethics

The following information was supplied relating to ethical approvals (i.e., approving body and any reference numbers):

Institutional Review Board and Ethics Committee of Lanzhou University Second Hospital.

## DNA Deposition

The following information was supplied regarding the deposition of DNA sequences:

The sequences are available at GEO: GSE86300, GSE184836, and at figshare: Zhao, Jing (2022): RNA-seq sequence of db/db mice. figshare. Dataset. https://doi.org/10.6084/m9.figshare.19823302.v2.

## Data Availability

The raw PCR data are available in the Supplemental File.

## Supplemental Information

Supplemental information for this article can be found online at http://dx.doi.org/10.7717/peerj.13932#supplemental-information.

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
