# Peer review of "Bioinformatics prediction and experimental verification of key biomarkers for diabetic kidney disease based on transcriptome sequencing in mice"

_PeerJ, doi:10.7717/peerj.13932_

## Round 0.1 · original submission · Major Revisions

When assessing your paper, the reviewers identified that the overall study is well designed. However, reviewers felt that additional details are needed to fully support the results and conclusions, and for the manuscript to be suitable for publication in this journal.

Reviewer 1 ·

Basic reporting

It is a well-written manuscript with sufficient introduction and background information. It is recommended that authors use proper gene nomenclature throughout the manuscript. In some places, the sentences are a little vague, for example lines 111-113. If the authors can clarify these points, it will make improve the readability of the manuscript. Figures legends need more description for better understanding.

Experimental design

Authors have defined the gap in knowledge and designed bioinformatics analysis to investigate the appropriate questions. Methods have been described with sufficient detail & information to replicate. Analysis of other datasets already in the literature provides additional robustness to the findings. However, there are no correlations with human disease highlighted. If authors can show relevance to the human disease, it will significantly elevate the findings presented here.

Validity of the findings

Confirmation of the findings with further functional studies will be necessary. Similarly, results from gene expression analysis should be confirmed at the protein level. Animal studies to validate these findings will be very important.

Reviewer 2 ·

Basic reporting

The manuscript is well written and organized.
It highlights the need for identification of biomarkers for early diagnosis of complications related to diabetic kidney disease.
However, the authors need to address following comments in the revision to explain and justify the impact of their study to the available literature.

Experimental design

1. C57 BLKS-db/db, a type 2 diabetes mice model was used for the current study. 6 weeks old mice was acclimatized for 20 weeks. what was the rationale behind choosing 6 week old mice and then feeding it for another 20 weeks?
2. Why did the authors choose only male mice? Was there a sex specific differential response?
3. Were the mice fed with normal or any particular diet for 20 weeks?
4. The study concentrates on identifying early markers for kidney disease in diabetic model. How does the study relate with early detection of DKD in these mice, which are diabetic and is 26 weeks old during the RNA isolation?

Validity of the findings

The results are extensive, shows changes in genes involved in lipid metabolism.

Additional comments

1. The authors have used mRNAs at few places (Page no: 8, Line 97 and 110) in the introduction, what do they infer? Do they mean gene expression ?
2. Similarly, in this line “We need to obtain more data by biotechnological methods and increase the number of experiments so that the results of……" (Page no: 8, Line 112) what biotechnological methods are required? It seems very generalized.
3. Did the authors happen to check blood pressure or have come across any literature which reports any changes in blood pressure?
4. Since the study reports changes in genes involved in lipid metabolism, is there any changes in the lymph transport or edema observed in these mice?

Reviewer 3 ·

Basic reporting

The authors have written this manuscript well, indicating the need for bioinformatic research in the field of DKD studies. The clarity of their narrative, results as well as method descriptions is high.

However, Figures 2 and 3 are unclear and thus cannot be read well. Authors will need to re-upload these figures for anyone to be able to understand the message through the data.

The following changes need to be made:
1. Line 147: Change “Biological replicates, biological replicates” to “Biological replicates. Biological replicates”. Ending the sentence makes sense in this case
2. Typo on Line 155: valid* in our experiment
3. How were the analyses mentioned in line 157- 160 performed? Authors should clarify these procedures
4. Line 162 seems to have an incomplete line ?” the expression matrix of FPKM can be obtained” did the authors intend to complete the sentence by adding “by…”
5. For plots described in section 2.2.2 and 2.2.3 authors should describe exact steps taken to visualize plots rather than just saying performed on website. This will aid in reproducibility by another method
6. 205 Student T- Test was employed for multiple comparisons – authors should state exactly the comparisons indicated
7. BP analysis? The authors do not mention the full form of BP – this makes it information dense for a new reader.
8. Line 223 has some typos authors should clarity what they mean by MF, “:” in middle of sentence etc.
9. Analysis scripts for Limma software were not provided. These are referenced in Figure 1 and should be added as supplementary documentation/ methods for this paper
10. In Figure 1, the text is blurry – authors should revise for clarity
11. Ref 31 seems to be missing a bracket at the end

Experimental design

The authors also explain how current treatments fail to find potential biomarkers, and early detection in DKD cases and demonstrate how their study will fill a knowledge gap. The authors also point out limitations of their study which will enable other studies to perhaps design similar studies better. The authors have received approval from an ethics board and documentation has been attached and was able to be downloaded.

Experimental methods are supplied with details to be able to be replicated. However, computational scripts are not shared and should be done so by the authors to enable replication and publication. Even figure generation done from a website should be described in detail - this has been pointed out in (1) citing exact lines.

Finally PPAR pathway seems to also be upregulated - the authors should comment on this finding in the discussion section.

Validity of the findings

The authors have used appropriate statistical tests and the studies are well controlled.

Additional comments

NA

---

## Round 0.2 · Minor Revisions

Thank you for making the changes. The reviewers are satisfied with most of your changes. However, Reviewer 2 has raised valid questions related to experimental design. Kindly address those questions.

Reviewer 1 ·

Basic reporting

Authors have addressed my comments satisfactorily.

Experimental design

Authors have addressed my comments satisfactorily.

Validity of the findings

Authors have addressed my comments satisfactorily.

Reviewer 3 ·

Basic reporting

Authors appropriately answered comments. They seem to have swapped multiple references out were these wrongly annotated initially? additional clarification needed.

Experimental design

The researches changed their report from six mice to twelve mice - this is a confusing change - does this indicate that the researchers did this study again or that they simply did not include this data in the first go. Additional clarification needed from authors on this.

Validity of the findings

The authors have not answered to sharing a git repository demonstrating their scripts.

---

## Round 0.3 · accepted · Accept

Thank you for submitting the manuscript.